# WILD COMPARISONS: A STUDY OF HOW REPRESENTATION SIMILARITY CHANGES WHEN INPUT DATA IS DRAWN FROM A SHIFTED DISTRIBUTION

**Davis Brown**[1], **Madelyn Shapiro**[1], **Alyson Bittner**[1], **Jackson Warley**[1], **Henry Kvinge**[1,2]
[1]Pacific Northwest National Laboratory
[2]Department of Mathematics, University of Washington

## ABSTRACT

Representation similarity functions, which apply geometric or statistical metrics to the hidden activations of two neural networks on a fixed set of input datapoints, have become important for assessing the extent to which models process and represent data in the same way. Past studies of these functions have largely been model-centric, focusing on varying the models under comparison while keeping the input data fixed. In particular, there have not been comprehensive evaluations of how the results change when either out-of-distribution inputs are used or when the data used in the comparison represents only a small part of the training set diversity (e.g., a single class). These are increasingly important questions as the community looks for tools to assess high-impact models which can be expected to encounter high-diversity, out-of-distribution data at deployment. In this work we explore the ways which input data affects the comparison of model representations. We provide evidence that while two model's similarity does change when either out-of-distribution data or data representing a subpopulation of the training set is used, relative changes (e.g., "model $A$ is more similar to $B$ than model $C$ is") are small for reasonable datasets. This robustness reinforces the idea that representation similarity functions are a promising tool for analysis of real-world models. Finally, we describe an intriguing depth-based pattern that arises in the representation similarity between different layers of the same model which could provide potential insight into how deep learning models process out-of-distribution data.

## 1 INTRODUCTION

In the last two years we have seen an increasing need for capable tools to audit, assess, and analyze deep learning models, either because their popularity and capability means that we can expect them to be deployed in many applications (e.g., foundation models) or because they specifically target high-consequence applications (e.g., medical imaging models). Model similarity functions are a promising tool for this task since they can be used to evaluate whether a new model processes and represents data in a way that is similar to models that may be better understood. For instance, when a new LLM is released, researchers might begin their assessment by comparing it to known LLMs and then use this information to guide downstream analysis.

To apply a representation similarity function, a researcher needs a pair of models and a collection of data whose representation (within each of the models) will be compared (see Figure 1). Existing research has focused on varying the models being compared. For instance, Ding et al. (2021) study whether a similarity function is sensitive to model's random initialization even when all other aspects of model architecture and training are held constant. On the other hand, little attention has been given to the data that is the basis of the model comparison. The standard procedure would be to assume that both models have the same training/test set and use examples from the test set as input to the similarity function. However, it is increasingly the case that some models (i) are expected to be applied to a broad set of data on deployment that might not be well-represented by the test set or (ii) who have a training set whose diversity is computationally infeasible to capture with current representation similarity functions.

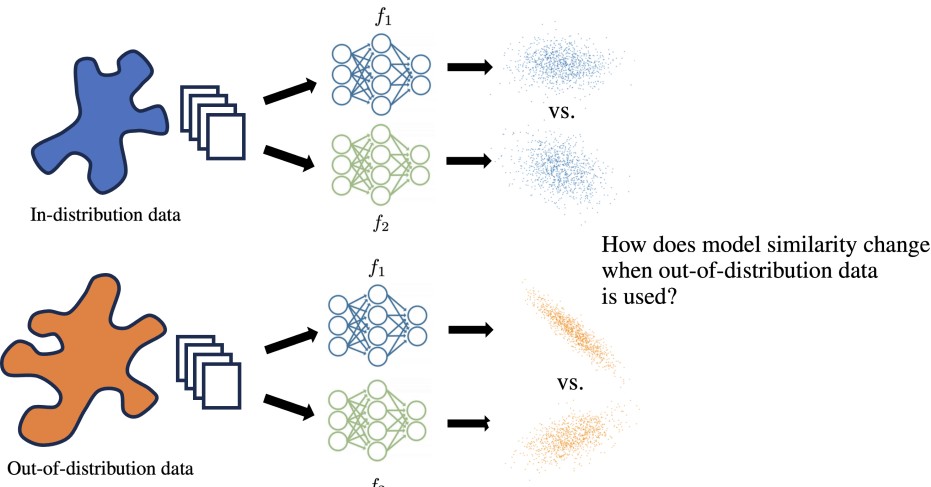

Figure 1: Past studies of representation similarity functions have used in-distribution data to evaluate model similarity. In this work we ask what happens when we do evaluation on out-of-distribution data.

Further, the question of not just **whether** two models are similar, but **where** they are similar is interesting from the perspective of the emerging science of deep learning. Indeed, from a mathematical point of view, a comparison of two functions $f_1$ and $f_2$ based on their value at a small number of points is fundamentally limited, particularly when $f_1$ and $f_2$ have the richness and complexity of modern deep learning models. For instance, it is well-known that representation similarity metrics can detect the changes to a model caused by adversarial training. Are these changes purely local to the data distribution where this training occurs, or do they extend to out-of-distribution data? A better understanding of these issues has the potential to help us understand and predict model behavior in the face of distribution shift, a fundamental challenge in real-world machine learning. Hence, we consider the following question:

> **Question:** *To what extent do model comparisons depend on the distribution we draw input data from?*

We describe an initial set of experiments to probe this question. We find that model similarities do change (sometimes substantially) as input data drifts further from the training distribution, but changes to relative similarity, by which we mean changes to the ranking (by similarity) between a fixed model $f_i$ and all other models under consideration $f_1, \ldots, f_k$, remains relatively minor for most datasets. Even datasets that can be characterized as coming from a different imaging domain, such as satellite images, induce only moderate changes in relative similarity. This is true when we control for differences in architecture, comparing only ResNet50 models with different training routines.

We see a similar phenomenon when we compare representations on subpopulations of the data. While the absolute similarity can change a surprising amount from subpopulation to subpopulation, relative similarity changes very little. These results, together with the above finding on out-of-distribution data suggest that **as long as one is only interested in the relative similarity of models, there is flexibility with the data that is used for comparison**. In terms of our understanding of deep learning models more generally, **our results suggest that changes to model representations induced by training (e.g., adversarial training, different augmentation schemes) have global effects, impacting the way a model represents data well outside its training distribution**.

Finally, besides looking at changes in representations between models, we also look at changes representations at different layers of the same model. Here we find a curious pattern in which early and late layers change substantially (relative to other layers) as we move to out-of-distribution

data, while changes to middle layers are substantially less. We speculate that this is due to OOD data having different input statistics and class outputs, but having largely similar features in the intermediate layers. Altogether our contributions in this paper include:

- Formulation of the question of how model representation similarity changes as a function of the input data distribution.

- Evidence that while absolute model similarity can change from data distribution to data distribution, the relative representation similarities of models can persist even when input data is taken out-of-distribution or from a subpopulation of the training distribution.

- We describe an interesting phenomenon where the representations at early and late layers of a model change more than middle layers when out-of-distribution input data is used.

## 2 RELATED WORK

There are a range of approaches to analyzing the hidden representations of deep learning models. Prominent examples not used in this work include canonical correlation (Hotelling, 1992) and its relatives such as SVCCA (Raghu et al., 2017) and PWCCA (Morcos et al., 2018). There are also topology-based similarity functions such as Geometry score (Khrulkov & Oseledets, 2018), representation topology divergence (Barannikov et al., 2021), or the similarity function described in (Purvine et al., 2023). Representations can also be compared via differences in statistics such as intrinsic dimension (Ansuini et al., 2019).

One set of similarity functions that are thematically related to this work are the neighborhood based similarity methods that use Jaccard distance (Schumacher et al., 2021), (Hryniowski & Wong, 2020), (Gwilliam & Shrivastava, 2022), cosine similarity (Hamilton et al., 2016), or rank similarity (Wang et al., 2020). Though we also consider relative distances in this work, our relative distances are between models rather than individual representations as in the papers above.

Representation similarity has been an important tool in better understanding how and why deep learning models work. For instance, Li et al. (2015) uses similarity functions to show that even with different initializations, models tend to learn the same features during training. Other works look at issues around the use of similarity functions. Sucholutsky et al. (2023) attempts to create a unified framework around representation similarity across machine learning, neuroscience, and cognitive science.

We a few works that also concentrate on the input data to similarity functions. Cui et al. (2022) explores a failure mode wherein the similarity or dissimilarity of the input data used confounds measurement of the similarity or dissimilarity of representations. In Section 4 we show that the similarity of models is higher for certain architectures when evaluating on subpopulations. This may be a consequence of the observation in Cui et al. (2022), as random images from a specific class may tend to be more similar than random images from all of ImageNet. However, we also show that these differences have little impact on relative similarity. Recently, Klabunde et al. (2023a) observe that the relative similarity between language models is dependent on the application of the models and, for example, the difficulty of the task. Finally, Gondhalekar et al. (2023) also looks at CKA values between different layers of the same model when out-of-distribution data is used as input. They find that different layers tend to be more similar in models that fail to generalize. This is distinct from our layer-based observation that the early and late layers of a model tend to change more (relative to other layers of the model) when any out-of-distribution data is applied.

## 3 BACKGROUND ON REPRESENTATION SIMILARITY FUNCTIONS

We briefly review representation similarity functions with a focus on those that we use in the experiments in this paper. For a comprehensive survey of this domain, see Klabunde et al. (2023b). Let $f_1, f_2 : X \rightarrow \mathbb{R}^n$ be neural networks with $f_i^{<k} : X \rightarrow Z_i^k$ the function defined by the first $k$ layers of $f_i$ where $i \in \{1, 2\}$. For instance, $f_1^{<10}$ is the first 10 layers of $f_1$ with $Z_1^{10}$ the space of hidden activations at layer 10. A *representation similarity function* is a function that quantifies the extent to which $f_1^{<k}(x_1), \ldots, f_1^{<k}(x_n)$ and $f_2^{<k}(x_1), \ldots, f_2^{<k}(x_n)$ are or are

not similar for some $x_1, \ldots, x_n \in X$. It is convenient to represent $f_1^{<k}(x_1), \ldots, f_1^{<k}(x_n)$ (respectively $f_2^{<k}(x_1), \ldots, f_2^{<k}(x_n)$) as a $n \times k_1$ matrix $A_1^{k_1}(x_1, \ldots, x_n)$ (resp. $n \times k_2$ matrix $A_2^{k_2}(x_1, \ldots, x_n)$) whose rows are the representation of each data point and whose columns are the $k_1$ (resp. $k_2$) dimensional features. In this context a representation similarity function $m$ takes $A_1^{k_1}(x_1, \ldots, x_n)$ and $A_2^{k_2}(x_1, \ldots, x_n)$ and returns a scalar value quantifying some aspect of the similarity of $A_1^{k_1}(x_1, \ldots, x_n)$ and $A_2^{k_2}(x_1, \ldots, x_n)$.

We next briefly review some of the representation similarity functions used in this work.

**Centered Kernel Alignment** (Kornblith et al., 2019): Centered Kernel Alignment (CKA) captures pairwise similarity within a given set of representations using a representation similarity matrix $[S]_{i,j}$. This is obtained by applying a kernel function to each pair of hidden activations $f_1^{<k}(x_i)$ and $f_2^{<k}(x_i)$ after they have been mean centered. The Hilbert-Schmidt Independence Criterion is then used to evaluate the extent to which to such similarity matrices $S_1, S_2$ are the same. This framework means that CKA is invariant to orthogonal transformation and does not require that the representations that are being compared have the same dimension.

**Orthogonal Procrustes Distance** (Ding et al., 2021): Aims to find orthogonal transformation $R$ from the group $O(d)$ where $d = \dim(Z_2^{k_2}) = \dim(Z_1^{k_1})$ such that the Frobenius norm $||A_1^{k_1}(x_1, \ldots, x_n) - A_2^{k_2}(x_1, \ldots, x_n)R||_F$ is minimized, i.e., can we make $A_2^{k_2}(x_1, \ldots, x_n)$ look like $A_1^{k_1}(x_1, \ldots, x_n)$ by rotating the latter? Orthogonal Procrustes is invariant to orthogonal transformations by construction.

**Permutation CKA and Procrustes Distance**: These two similarity functions are analogous to those above but instead of being invariant to any orthogonal rotation in the activation basis, they are only invariant to permutation of activations. The reasoning for this comes from the fact that multiple works have either used theory (Godfrey et al., 2022) or empirical observation (Entezari et al., 2021),(Bau et al., 2017) as support for the idea that permutations (rather than any orthogonal transformation) are the family of transformations that hidden representations should be invariant to. We refer the reader to Godfrey et al. (2022) for more details on permutation CKA and Williams et al. (2021) for more details on permutation Procrustes.

## 4 EXPERIMENTS

**Do relative similarities among models persist when out-of-distribution data is used?:** Recall that one of the questions that we are interested in is the extent to which relative similarity changes when out-of-distribution data is used as input to a representation similarity function. We focus our experiments on a collection of models that have fixed architecture and primarily differ in the type of training schemes used, as we would like to know whether relative similarity induced when training on certain distributions persists when measured on other distributions. We restrict ourselves to comparing latent representations for this initial study.

Our set of models consists of a range of different ResNets, all trained on ImageNet (Russakovsky et al., 2015), including torchvision pretrained ResNets (Marcel & Rodriguez, 2010) with both 'version 1' and 'version 2' weights (the latter with more sophisticated augmentation schemes, learning r ate scheduler, etc.). We also include several ResNet50 models trained with different types of heavy augmentation (e.g., PRIME (Modas et al., 2022), deepaugment (Hendrycks et al., 2021a), augmix (Hendrycks et al., 2019)) or data with style transfer applied (Geirhos et al., 2018), as well as 8 different ResNet50's trained with adversarial training for varying $\epsilon$ values (Salman et al., 2020). We summarize these models in Section A.1 of the Appendix.

We then choose five datasets that are distinct from ImageNet: vehicle chips from the satellite imaging dataset xView (Lam et al., 2018), ImageNet-R (Hendrycks et al., 2021a), MNIST Deng (2012), ObjectNet (Barbu et al., 2019), and Describable Textures (Cimpoi et al., 2014). For each of these datasets $D_k$ we (i) choose a random set of 500 images $X_k$, (ii) use these images as input to each pair of models $(f_i, f_j)$, (iii) extract the corresponding latent representations $f_{i,\text{latent}}(X_k)$ and $f_{j,\text{latent}}(X_k)$, and (iv) apply CKA and permutation CKA to each of these pairs $f_{i,\text{latent}}(X_k)$ and $f_{j,\text{latent}}(X_k)$ to get $\kappa_{i,j,k}^{\text{CKA}}$ and $\kappa_{i,j,k}^{\text{pCKA}}$. Note that each $\kappa_{i,j,k}^{\text{CKA}}$ (respectively $\kappa_{i,j,k}^{\text{pCKA}}$) depends on the pair of models (indices $i$ and $k$) and the dataset (index $k$).

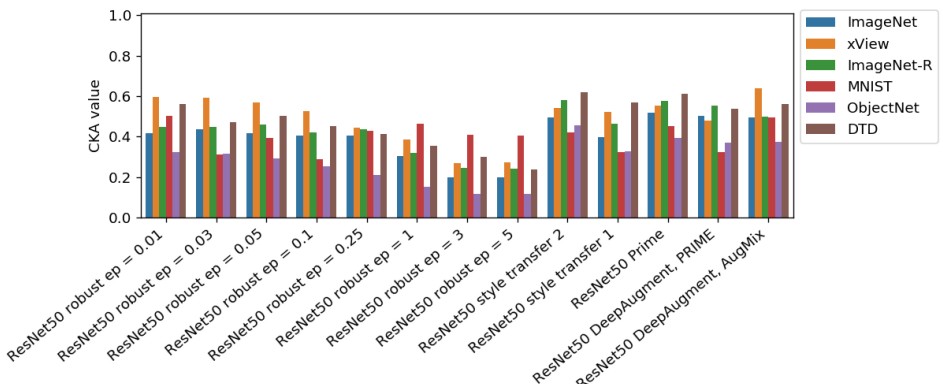

Figure 2: The latent representation CKA similarity ($y$-axis) between a ResNet50 trained on ImageNet (using the 'version 1' pretrained weights from torchvision) and a range of other ResNet models ($x$-axis), all trained on ImageNet. Colors indicate the dataset that input data was drawn from.

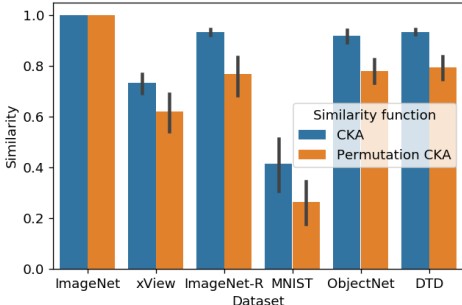

Figure 3: Spearman's rank correlation coefficient ($y$-axis) captures the changes in relative similarity when two models are evaluated on their test data (ImageNet) vs OOD datasets ($x$-axis) using CKA or permutation CKA. Larger values (with a maximum of 1) indicate that the relative similarity between models is preserved when we evaluate them on a given OOD dataset. A value of $0$ would mean there is no correlation between relative similarity on ImageNet vs OOD data and a value of $-1$ would mean that similarity is exactly reversed on OOD data.

In Figure 2, we plot CKA values for the latent representations of a torchvision ResNet50 model trained on ImageNet (with 'version 1' pretrained weights) and range of other ResNet50 models ($x$-axis). The different colored bars indicate the dataset from which input data was drawn. We see that CKA values can change substantially when moving from ImageNet (blue bars) to out-of-distribution datasets. There are some patterns in these changes. For instance, models tend to be more similar when data from xView, DTD, or ImageNet-R is used, while they are less similar on ObjectNet. Whether similarity increases on MNIST depends on the model that one uses. We speculate that this pattern may relate to the complexity or semantic variety of the datasets. xView images are low-resolution and have low-semantic variety (being mostly low-resolution chips of satellite images with blurry vehicles. ImageNet-R images are drawings, paintings, etc. and hence tend to have less complexity, and DTD is textures with repeated patterns. On the other hand, ObjectNet has high complexity images with clutter. See Figure 4 for examples. MNIST images are simple, but they may be far enough out-of-distribution to not follow the pattern.

To understand how the **relative** similarity between models changes as we move to out-of-distribution datasets, for each dataset $k$ and model $f_i$, we construct a vector of similarities (for CKA or permutation CKA) between $f_i$ and all of the other models ($\{\kappa_{i,j,k}^{\mathrm{CKA}}\}_j$ or $\{\kappa_{i,j,k}^{\mathrm{CKA}}\}_j$). Then for each $f_i$, we compare the vector measured on ImageNet with the vector measured on one of the out-of-distribution datasets using Spearman's rank correlation coefficient. If this value is close to 1, it means that the

ranking of similarities of $f_i$ to other models does not change as we move out-of-distribution, if it is 0, it means there is no correlation between both rankings, and if it is $-1$ it means that one ranking is the reverse of the other.

We show the result of this experiment in Figure 3. While CKA and permutation CKA values change when out-of-distribution data is used as input, we see that changes in relative similarity are minor. This is the case even when all models have similar architectures and were trained on the same dataset (albeit with different augmentation schemes). The only dataset where relative similarity changes substantially is MNIST, which we would argue is significantly out of distribution.

We see similar results when we perform our measurements using Procrustes distance and permutation Procrustes (Figure 9). We note that the two similarity functions that are only permutation invariant tend to be more impacted by changes in distribution. We suspect that this is due to the fact that such similarity functions are more sensitive to changes in semantic content given the importance of the activation basis in this context (Godfrey et al., 2022).

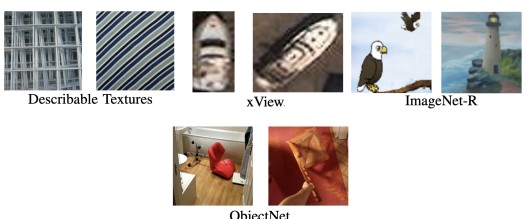

Figure 4: Examples of the OOD datasets that we use in our experiments.

> **Takeaways**
>
> - CKA values do change, sometimes substantially, when out-of-distribution data is used. The direction of these change may relate to properties of the dataset.
> - While representation similarity values change when out-of-distribution data is used, changes to the relative similarity between models tend to be minor.

**What do these results say about a model's representations?**: Our experiments support the idea that relative similarity is persistent well-beyond the data distribution where a model is trained. We interpret this as evidence that the way that a model is trained has a substantial impact on how it represents data even when we move outside of the training distribution. While this may not seem surprising at first glance, specifying the value of an arbitrary function at a finite number of points tells us very little about how that function will behave elsewhere, especially in the cases that we consider where the support of these points (the training distribution) is a low-dimensional manifold that does not even take up a sliver of the high-dimensional input space. So just as different neural networks, that could easily learn different representations of data and still solve a problem instead often learn similar representations (Li et al., 2015), a neural network that in theory should be able to have certain properties on the training distribution and different properties elsewhere ends up being defined largely by learning that is done on the thin slice of input space where the training data sits.

**Do relative similarities persist when we draw data from a subpopulation of the training distribution?**:

We ask a similar question to that above but for subpopulations of the training distribution rather than out-of-distribution data. For instance, one might worry that model $A$ would be measured to be more similar to model $B$ than model $C$ when we sample from one class but $A$ would be measured as more similar to $C$ when we use data from another class. To test this we pick 5 random classes from ImageNet: Scorpion, Ant, Otter, Barbell, and Pay-phone. Restricting our input data to each of these in turn, we run CKA on the same pairs of ResNets from the experiment above. We plot CKA values between a ResNet50. Then, as above, we construct vectors with ranked similarity for each fixed model. Using Spearman's rank correlation coefficient we compare rankings corresponding to all of ImageNet and rankings corresponding to a single class. We show our result in Figure 6, where we again find that even though model similarity shows variance between different classes, changes to relative similarity are minor. Of course, classes are only one way to partition a dataset. It would be

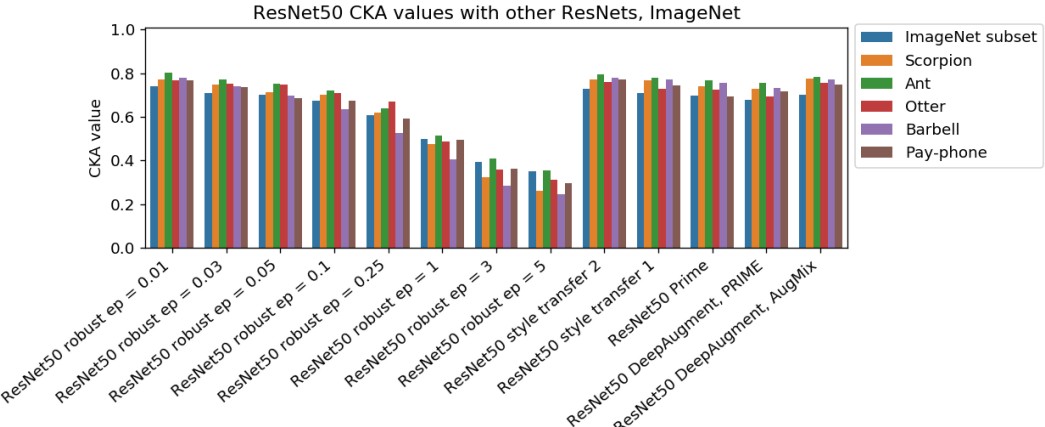

Figure 5: The latent representation CKA similarity ($y$-axis) between a ResNet50 trained on ImageNet (using the 'version 1' pretrained weights from torchvision) and a range of other ResNet models ($x$-axis), all trained on ImageNet. Colors indicate the class that input data was drawn from.

interesting to understand the effect of restricting to different types of subpopulations (e.g., different classes but same background, same low-level pixel statistics, etc.).

We see similar results when we perform our measurements using Procrustes distance and permutation Procrustes (Figure 9)

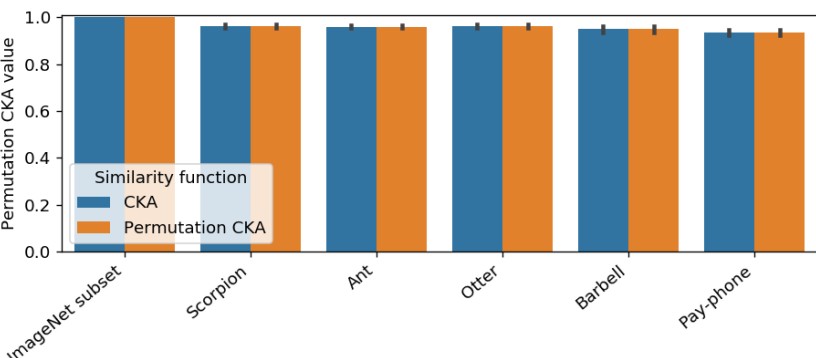

Figure 6: Spearman's rank correlation coefficient ($y$-axis) captures the changes in relative similarity when models are evaluated on a combination of 5 different classes of ImageNet vs a single one of those classes ($x$-axis). Larger values (with a maximum of 1) indicate that the relative similarity between models is preserved when we restrict their evaluation to a single class. Black bars indicate 95% confidence intervals.

> **Takeaways**
>
> Limiting the input data of a representation similarity function to a single class has a small effect on the relative similarity of models.

**Do the representations *within* the layers of a model remain consistent when OOD data is used?**:

We also examine how the representations between layers in a *single model* change when evaluated on OOD data. We find that representations at very early and very late layers change the most (with respect to the representations found at other layers). Modifying the notation used above we write $\kappa_{i,j,k}^{\text{CKA},\ell}$ (resp. $\kappa_{i,j,k}^{\text{pCKA},\ell_1,\ell_2}$) for the CKA (resp. permutation CKA) value for model $f_i$ and $f_j$, on dataset $k$, at layer $\ell_1$ in $f_1$ and $\ell_2$ in $f_2$. Thus, when $i = j$, we can form the matrix which consists of CKA values for each pair of layers $\ell_1$ and $\ell_2$ in $f_i$, $K_{i,k}^{\text{CKA}} := [\kappa_{i,i,k}^{\text{pCKA},\ell_1,\ell_2}]_{\ell_1,\ell_2}$. To study how

this matrix changes when out-of-distribution data is used as input, we explore the representations of `timm` models (Wightman, 2019) on ImageNet data (the training distribution) and ImageNetA (Hendrycks et al., 2021b), ImageNetV2 (Recht et al., 2019), ImageNet-R, and ObjectNet (out-of-distribution data).

We ask how representations change with respect to depth– that is, which layers change the most as we move out-of-distribution? To measure this, we first take the difference between $K_{i,\text{ImageNet}}^{\text{CKA}}$ and $K_{i,k}^{\text{CKA}}$, which tells us the absolute change from ImageNet and an OOD dataset $k$ in between-layer-similarities. The norm over each row $i$ gives the absolute change in how similar a layer $i$ is to other layers $j$, which we normalize to get the relative change:

$$\frac{\|K_{i,\text{ImageNet}}^{\text{CKA}} - K_{i,k}^{\text{CKA}}\|_2^{(i)}}{\|K_{i,\text{ImageNet}}^{\text{CKA}}\|_2^{(i)}}. \tag{1}$$

We compute this metric for the OOD datasets $k$, across 428 `timm` models. Results for ImageNetV2 are shown in Figure 7, where we bin across relative depth (which varies from 0 to 1 based on the fraction of layers the data has passed through). We find that early and late layers show systematically larger differences on OOD data than middle layers. We include similarity heatmaps for a random subset of models along with the other datasets, which show nearly identical trends, in the Appendix. We speculate that the image-level statistics are different between in-distribution and out-of-distribution data, causing representations close to the input to be different. On the other hand, we know that later layers capture much of the class output differences, so these may be different because models are less accurate /class representations are different on OOD data. Further experiments would be necessary to validate these hypotheses.

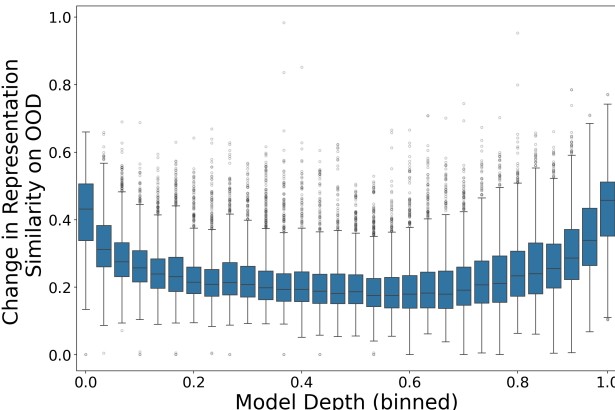

Figure 7: The relative changes in similarity scores with model depth between ImageNet and ImageNetV2. Early and later layers show systematically larger similarity differences.

> **Takeaways**
>
> Early and late layers of a model change more with respect to the other layers on OOD data– that is, their relative representation similarities shift more prominently than middle layers.

## 5 CONCLUSION

In this paper we explore how representation similarity values change when either out-of-distribution data or subpopulations of the training data are used as input. We find that while similarity does change in these situations, relative similarity does not, meaning that models that are similar on in-distribution data tend to be similar on out-of-distribution data. This not only provides evidence that model comparisons that hold in the lab, often persist when those models are deployed and encounter data from a shifted distribution but also that while training is performed on a specific distribution

(usually a thin slice of the input space), the impact of training extends its imprint far beyond to out-of-distribution data.

ACKNOWLEDGMENTS

This research was supported by the Mathematics for Artificial Reasoning in Science (MARS) initiative at Pacific Northwest National Laboratory. It was conducted under the Laboratory Directed Research and Development (LDRD) Program at at Pacific Northwest National Laboratory (PNNL), a multiprogram National Laboratory operated by Battelle Memorial Institute for the U.S. Department of Energy under Contract DE-AC05-76RL01830.

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

# A  APPENDIX

## A.1  RESNET MODELS USED IN EXPERIMENTS

Several of our experiments in Section 4 use a collection of ResNet models. We provide a list of these models in Table 1.

## A.2  REPRESENTATION SIMILARITIES BETWEEN THE LAYERS OF THE SAME MODEL

Table 1: The collection of ResNet models used in our experiments.

| Augmentation | Notes |
|---|---|
| ResNet18 | |
| ResNet34 | |
| ResNet50 | Pretrained weights v1 |
| ResNet50 v | Pretrained weights v2 |
| ResNet101 | Pretrained weights v1 |
| ResNet101 | Pretrained weights v2 |
| ResNet152 | Pretrained weights v2 |
| ResNet152 | Pretrained weights v2 |
| WideResNet50 | Pretrained weights v1 |
| WideResNet50 | Pretrained weights v2 |
| WideResNet101 | Pretrained weights v1 |
| WideResNet101 | Pretrained weights v2 |
| ResNet50 | adversarial training $\epsilon = 0.01$ |
| ResNet50 | adversarial training $\epsilon = 0.03$ |
| ResNet50 | adversarial training $\epsilon = 0.05$ |
| ResNet50 | adversarial training $\epsilon = 0.1$ |
| ResNet50 | adversarial training $\epsilon = 0.25$ |
| ResNet50 | adversarial training $\epsilon = 1$ |
| ResNet50 | adversarial training $\epsilon = 3$ |
| ResNet50 | adversarial training $\epsilon = 5$ |
| ResNet50 | style transfer augmentation v1 |
| ResNet50 | style transfer augmentation v2 |
| ResNet50 | PRIME augmentation |
| ResNet50 | deepaugment and PRIME augmentation |
| ResNet50 | deepaugment and augmix |

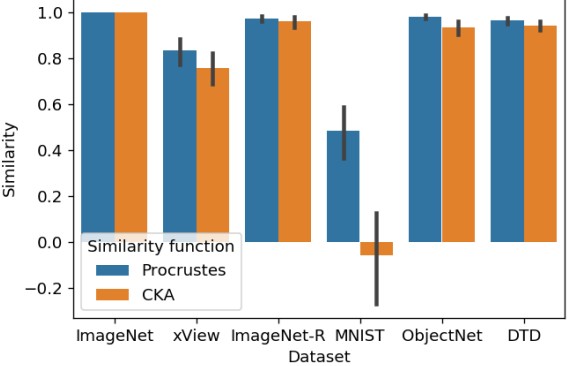

Figure 8: Spearman's rank correlation coefficient ($y$-axis) captures the changes in relative similarity when two models are evaluated on their test data (ImageNet) vs OOD datasets ($x$-axis) using Procrustes or permutation Procrustes. Larger values (with a maximum of 1) indicate that the relative similarity between models is preserved when we evaluate them on a given OOD dataset. A value of 0 would mean there is no correlation between relative similarity on ImageNet vs OOD data and a value of $-1$ would mean that similarity is exactly reversed on OOD data.

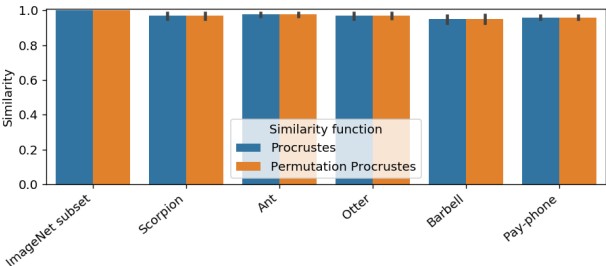

Figure 9: Spearman's rank correlation coefficient ($y$-axis) captures the changes in relative similarity using Procrustes and permutation Procrustes when models are evaluated on a combination of 5 different classes of ImageNet vs a single one of those classes ($x$-axis). Larger values (with a maximum of 1) indicate that the relative similarity between models is preserved when we restrict their evaluation to a single class. Black bars indicate $95\%$ confidence intervals.

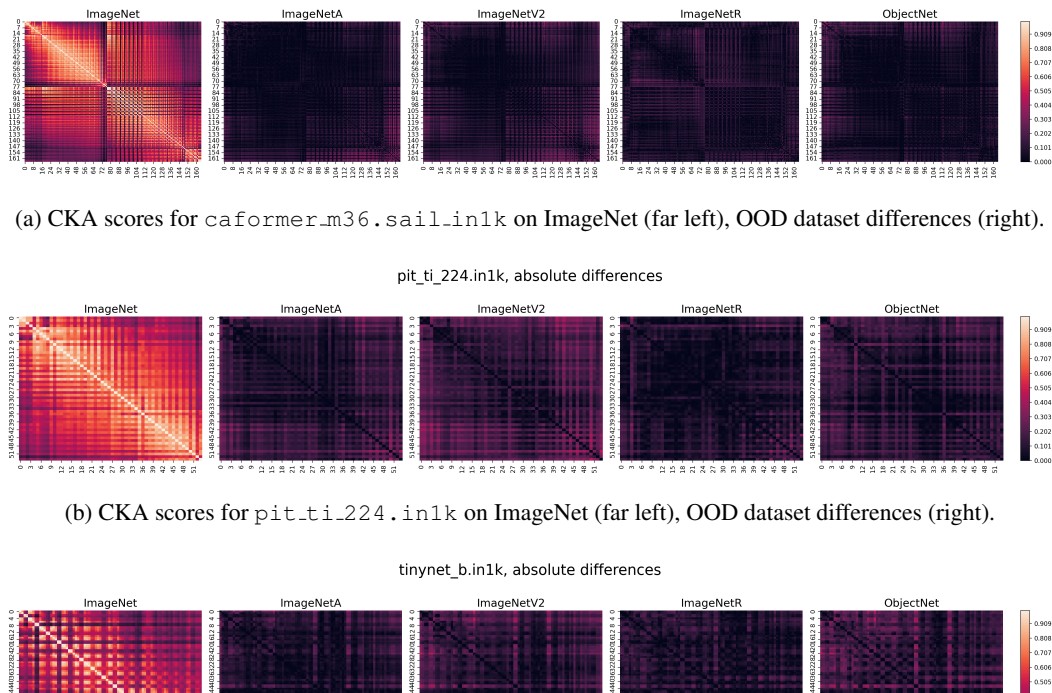

(a) CKA scores for `caformer_m36.sail_in1k` on ImageNet (far left), OOD dataset differences (right).

(b) CKA scores for `pit_ti_224.in1k` on ImageNet (far left), OOD dataset differences (right).

(c) ImageNet CKA scores for `tinynet_b.in1k` on ImageNet (far left), OOD dataset differences (right).

Figure 10: CKA scores for ImageNet (far left), and the absolute value differences between ImageNet and OOD datasets, for a few `timm` models.

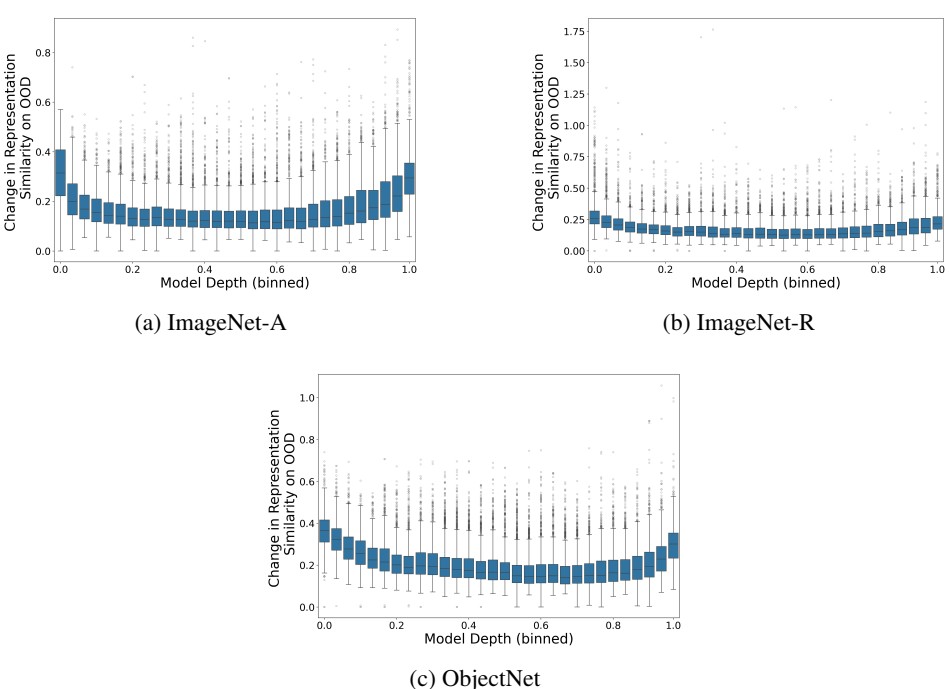

(a) ImageNet-A

(b) ImageNet-R

(c) ObjectNet

Figure 11: CKA scores for ImageNet (far left), and the absolute value differences between ImageNet and OOD datasets, for a few `timm` models.

