# OpenReview forum: "Wild Comparisons: A Study of how Representation Similarity Changes when Input Data is Drawn from a Shifted Distribution"
_ICLR.cc/2024/Workshop/Re-Align — ICLR 2024 Workshop Re-Align Poster_

### Official Review · Reviewer_DHrB · 2024-02-19
**I propose acceptance, as the paper generates interesting insights.**

**Rating:** 2
**Fit:** 3
**Confidence:** 3

**Workshop Review:**

*Summary:*
- The paper investigates the similarity between model representations when input data is out of distribution.
- The main contributions of the paper are its novel perspective from inspecting out of distribution data and relative similarity (similarity by ranking).

*Strengths:*
- The paper takes a novel perspective by evaluating on out-of-distribution data and quantifying relative similarity.
- The 3 experiments defined are interesting and generate good insights.
- I believe the insights/take is interesting for the wider community and will inspire further research.

*Weaknesses:*
- Some parts of the paper are very hard to read.
- I believe the 'Experiments' section would benefit from a more traditional and clear structure, such as models, datasets, and the experimental setup.
- The paper lacks an investigation of where the dissimilarities emerge from, allowing for deeper insights.
- The paper only investigates ResNet and a limited set of datasets.

*Recommendations:*
- Accept (Poster)
- I recommend accepting this paper as it presents an interesting investigation into model representations.
- I recommend accepting for a poster, as I believe the conclusions that can be drawn from the results are not clear enough for a talk.

*Supporting Arguments for Recommendation*:
- Within the setting of the workshop, I think the paper presents enough novel results to warrant acceptance.
- I believe the paper needs some restructuring/rewriting and more experiments (read more models) to move towards conference publication level.

*Questions:*
- Can you quantify the changes to relative similarity in a more fine-grained manner? What I am wondering is how close the ranking is to changing.
- Would it be possible to have more controlled toy experiments to check where dissimilarity in out-of-distribution data representations comes from? What I am trying to get at is whether you can make more assertive statements about where the trends emerge from, rather than speculating on dataset similarity/complexity.

*Feedback:*
- I would recommend rereading the paper aloud and shortening sentences. For example, the second sentence of the fourth paragraph in the introduction is very hard to read, but highly important because it introduces relative similarity.
- I think Figure 1 can be made smaller, removed or needs to be improved. While it communicates what the papers research question is, it is not useful beyond this.
-  I think it would also be interesting to investigate how similarity changes within layer with increasing width. I know that this is harder to do, as the representations will be different in dimensionality, but maybe something to think about.

**Reason For Not Giving Higher Score:**

I did not recommend the paper for a talk, as the results are not substantial enough or allow clear conclusions (e.g. where dissimilarity comes from in the OOD data) to fill a talk.

**Reason For Not Giving Lower Score:**

I believe the paper is a very good start and interesting to the wider community (e.g. inspiring future research). Therefore, I believe it does not warrant a lower score.

**Reviewer Domain:**

machine learning

---

### Official Review · Reviewer_8uPr · 2024-02-22
**Interesting but perhaps not very novel**

**Rating:** 2
**Fit:** 3
**Confidence:** 2

**Workshop Review:**

The paper investigates how model representation similarity changes depending on the input distribution. They find that while changes to absolute similarity are substantial, the relative ordering of models in their similarities does not change.

**Clarity**

The paper is fine to read but I found the writing unintuitive at parts: for instance, in the abstract it says _relative changes (e.g., ''model A is more similar to B than model C is'') are small for reasonable datasets_. I found this confusing and I think something like _changes in the relative similarity between models (e.g., ''model A is more similar to B than model C is'') are small_ would be easier to understand instead. Also, what is a reasonable data set here?

**Correctness**

The methods seem correct and the results are about what I would expect.

**Novelty**

While the framing of the investigation is interesting, I think similar work has already been done: Supplementary C.3, Table 9 in [1] calculated CKA on OOD data over layers and also find that early and late layers have a lower rank correlation than later layers. Supplementary D, Figure 10 in [2] shows CKA over layers for both in- and out-of-domain data. Also [3] show in Supplementary I that odd-one out agreements reflect representational similarity and they investigate odd-one out agreements on OOD data. Ideally, the authors could add a section on how these studies relate to their results.

**Relevance for community**

Even though I don't consider the results to be particularly novel, I do think the work fits well within the scope of the workshop and is still of interest to the community.


[1] Ding, Frances, Jean-Stanislas Denain, and Jacob Steinhardt. "Grounding representation similarity with statistical testing." arXiv preprint arXiv:2108.01661 (2021).

[2] Mathis, Alexander, et al. "Pretraining boosts out-of-domain robustness for pose estimation." _Proceedings of the IEEE/CVF Winter Conference on Applications of Computer Vision_. 2021.

[3] Muttenthaler, Lukas, et al. "Human alignment of neural network representations." _arXiv preprint arXiv:2211.01201_ (2022).

**Reason For Not Giving Higher Score:**

I outlined some of the reasons in **Clarity** and **Novelty** above. Other minor points include:

- I kind of stumbled over the claim that changes to relative similarity are minor? In Figure 3 the rank correlation is up to 0.4 lower for xView compared to ImageNet.
- Might want to consider using a perceptually uniform colormap for Figure 2.
- Why is there a question mark at the end of the sentence describing orthogonal procrustes distance?
- Headline A.2 is displaced?
- Figure 10 in the Appendix, I'm assuming these are the layer wise model similarity heatmaps? X and Y labels would be nice to make this more intuitive. In general the Figure is very small.
- Figure 11 caption is the same as for Figure 10 and likely wrong?

**Reason For Not Giving Lower Score:**

In general the underlying question is interesting: _do representation similarity metrics remain a valid tool for model comparisons even on OOD data_? Also, it fits well within the scope of the workshop and is likely still of some interest to the community.

**Reviewer Domain:**

machine learning

---

### Official Review · Reviewer_pxgc · 2024-02-23
**Important research question for the representation comparison field, clear experiments**

**Rating:** 3
**Fit:** 3
**Confidence:** 3

**Workshop Review:**

**Summary:**

The paper is primarily concerned with the question “To what extent do model comparisons depend on the distribution we draw input data from?” They consider two extreme cases, one where the input data is drawn from a subset of the “in-distribution” and one where the input data is out-of-distribution. They consider ResNet models trained on ImageNet for their experiments and the CKA, permutation CKA, Procrustes distance and permutation Procrustes representation comparison methods. Contributions: while absolute measures of representation similarity can vary widely for OOD data (less so for subset of the training distribution), the relative similiarities (i.e. “model A is closer to model B than to model C”) is relatively robust to the input data. They highlight an interesting phenomenon where OOD data representations at early and late layers change more than for middle layers.

See "Weak points" and "Strong points" for the actual main comments.

**Questions/Feedback/Suggestions:**

- Which representations of the ResNet models are compared in Figures 2 and 3, at which layers of the networks are we comparing the representations? I assume it’s the same layer from two different networks being compared since this is standard, but is this averaged over all possible layers or only one layer is taken? A split of this analysis into “early”, “mid” and “late” layers representation comparison would be interesting.
- “The only dataset where relative similarity changes substantially is MNIST, which we would argue is significantly out of distribution.” → It would be interesting to see if we could have a ranking of the “OOD-ness” of the datasets as well and see if higher drops in similarity is correlated with being more out of distribution.
- What is the difference between the ResNet models used early in the paper and the timm ones mentioned later? This isn’t quite clear.
- There is no reference to Figure 5 in the text.

**Typos:**

- “We a few works…”  beginning of the last paragraph of section 2
- Procrustes results mentioned in OOD section point to subset results (fig 9 but it should be fig 8 I think)

**Reason For Not Giving Higher Score:**

**Weak points:**
- Representation comparison is still an active area of research and to the best of my knowledge there is no single “correct” representation similarity measure to use. Therefore, I would also suggest using PWCCA, nonlinear kernel CKA, and possibly some of the other options discussed in [1]. Also, it would be important to discuss the results obtained by the different representation similarity measures in light of their different invariances and sensitivities as suggested by [2].
- Procrustes results should be included in the main paper, same for any other representation similarity measure considered.
- Missing multiple citations in the introduction.
- Maybe links with theory would be interesting.

**References:**

[1] Harvey et al., UniReps at NeurIPS 2023, Duality of Bures and Shape Distances with Implications for Comparing Neural Representations

[2] Davari et al., ICLR 2023, Reliability of CKA as a Similarity Measure in Deep Learning

**Reason For Not Giving Lower Score:**

**Strong points:**
- The question of interest is highly relevant. So far representation similarity measures have mainly been applied to in-distribution data points. However, modern models are increasingly expected to be robust to large variations in input data distribution. Therefore, better understanding how representation similarity measures behave when OOD data is used as input is an important research direction.
- I like the fact that the authors consider both the local case (subset of the “in” distribution) and the OOD case.
- The relative change of similarity scores is an interesting analysis tool.
- Generally well written and clear text (although maybe a bit repetitive in places).

**Reviewer Domain:**

machine learning

---

### Decision · Program_Chairs · 2024-03-02

Accept (Poster)